# Model Interpretation Considering Both Time and Frequency Axes Given Time Series Data

**Woonghee Lee** [1], **Gayeon Kim** [2], **Jeonghyeon Yu** [2] and **Younghoon Kim** [2,*]

1. Department of Computer Science & Engineering, Hanyang University, Ansan 15588, Republic of Korea
2. Major in Bio Artificial Intelligence, Department of Applied Artificial Intelligence, Hanyang University, Ansan 15588, Republic of Korea
* Correspondence: nongaussian@hanyang.ac.kr

**Abstract:** Recently, deep learning-based models have emerged in the medical domain. Although those models achieve high performance, it is difficult to directly apply them in practice. Specifically, most models are not considered reliable yet, while they are not interpretable. Therefore, researchers attempt to interpret their own deep learning applications. However, the interpretation is task specific or only appropriate for image data such as computed tomography (CT) scans and magnetic resonance imaging (MRI). Currently, few works focus on the model interpretation given time series data such as electroencephalography (EEG) and electrocardiography (ECG) using LIME. Because the explanation generated by LIME is from the permutation of the divided input data, the performance of interpretation is highly dependent on the split method. In the medical domain, for the time series data, existing interpretations consider only the time axis, whereas physicians take account of the frequency too. In this work, we propose the model interpretation using LIME considering both time and frequency axes. Our key idea is that we divide the input signal using graph-based image clustering after transforming it using short-time Fourier transform, which is utilized to capture the change of frequency content over time. In our experiments, we utilize real-world data, which is EEG signals recorded from patients during polysomnographic (PSG) studies, as well as prove that ours captures a significantly more critical explanation than the state-of-the-art. In addition, we show that the representation obtained by ours reflects the physician's standard such as K-complexes and delta waves, which are considered strong evidence of the second sleep stage and a clue of the third sleep stage. We expect that our work can be applied to establish computer-aided diagnosis as well as to measure the reliability of deep learning models taking the time series into them.

**Keywords:** model interpretation; electroencephalogram; short-term Fourier transform

## 1. Introduction

Recently, deep learning-based models have been tremendously developed by many researchers in the medical domain for image and signal datasets. Specifically, for image data such as computed tomography (CT) scan, magnetic resonance imaging (MRI), deep learning models based on U-Net [1], which is proposed to segment biomedical images, achieve high performance. The studies include organ segmentation, brain tumor detection and skin cancer classification [2–6]. In addition, deep learning models are applied to one-dimensional medical data as well. The works utilize convolutional neural networks (CNNs) and long short-term memory networks (LSTMs) for arrhythmia detection, mortality prediction, sleep stage detection, bruxism detection and insomnia detection [7–14]. However, those methods are difficult to directly apply in practice. The reason why is that the deep learning-based model is hard to interpret. The lack of interpretability means the lack of reliability. Although, some works attempt to interpret the prediction results, the interpretation highly depends on their specific task [15–17].

Since the interpretation methods for deep learning have been proposed, such as Grad-CAM [18] the gradient-based visualization, KernelSHAP [19] the Shapley value-based method from game theory and LIME, which exploits surrogate model [20], the interpretations are also applied to the medical domain [21–23]. However, Grad-CAM-based visualization is only appropriate for image data. Besides, the exact Shapley value is difficult to earn, because the Shapley value is calculated from all combinations of the features [19]. Therefore, for one-dimensional data, LIME is utilized in the state-of-the-art interpretations, which are named NEVES [24] and LIMESegment [25].

Given a target model that is to be explained and the input data, LIME exploits a surrogate model that is considered easy to interpret, such as linear model or decision tree [20]. The explanation is obtained from training the surrogate model using perturbed samples and associated logits predicted by the target model. The perturbed samples are a randomly permuted subset of the input. For example, given an image of skin cancer, the image can be divided into lesions and normal skin of the image. Then, the perturbed samples can be the image of lesions, the image of normal skin and the image of both. Because the characteristics of LIME, the explanation depends on how to split the input instance. Former works exploit LIME, split the input time series at the same interval and fill it with interpolation [24] or divide from the change point, which is the boundary of the most dissimilar between window slides [25]. They are, however, limited in that they consider the signal changing along the time axis whereas physicians consider the characteristics of signals along the time and frequency axes in practice [26–29].

In this work, we propose the method to interpret models based on deep learning given time series medical data. Specifically, we consider the medical signals, which are that not only the change point along the time axis but also the frequency plays an important role in diagnoses such as electroencephalography (EEG), electromyography (EMG) and electrocardiography (ECG) [30–32]. Given the input signal, our suggestion considers both time and frequency axes. We transform the input signal using short-time Fourier transform (STFT) and divide it into aspects of both axes. The divided signal is randomly permuted and inverted by inverse STFT (ISTFT). Finally, our method produces the explanation of the input as depicted in Figure 1. In the experiments, we show that our method figures out a significantly more critical explanation than the state-of-the-art and it generates a comparative explanation as a physician's decision.

Our contributions are as follows:

- We propose the novel approach to interpret a deep learning classifier given medical signals such as EEG considering both aspects of time and frequency while physicians consider both of them to diagnose in practice.
- In the experiments, we confirm the suggestion from this work with the real-world dataset, which is EEG signals recorded from patients during polysomnogrphic studies.
- We show that our suggestion captures the probable explanations such as K-complexes and delta waves, which are considered strong evidence of the second sleep stage and the third sleep stage, respectively.

The rest of our paper is organized as follows. In Section 2, we review existing interpretations for deep learning models and their limitations in the general and medical domains. We define our problem and introduce our method in Section 3. We compare the performance between ours and the state-of-the-art in Section 4. Finally, we conclude our research in Section 6. Moreover, we attach additional experimental results in Appendix A.

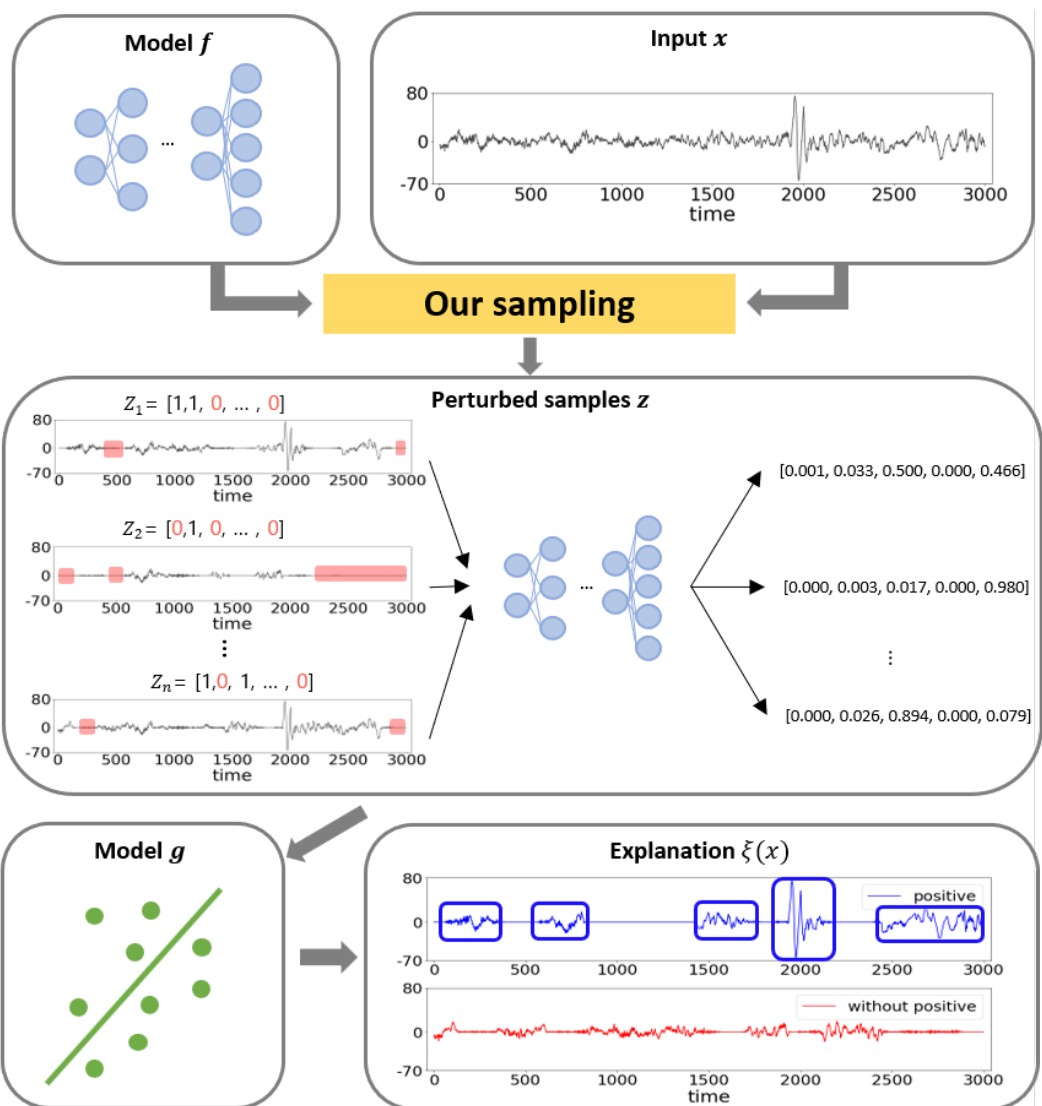

**Figure 1.** Given the target model $f$, which is to be explained, and the input signal $x$, which is an EEG signal, the explanation $\xi(x)$ is obtained using the surrogate model $g$, which is easily interpretable as a linear model. The model explanation $\xi(x)$ depends on the perturbed samples $z$.

## 2. Related Work

Although deep learning model interpretation has been paid attention by many researchers [18,19,33], those works are limited to the task for image data. Some works consider the medical domain, but most studies consider only image data such as computerized tomography (CT) scan and skin lesions [15–17], although only a few works attempt to interpret models given the time series data [24,25]. In this section, we review existing deep learning model interpretation methods in a general domain and in a specific domain, which is the medical domain, as well as the model interpretation given time series data.

### 2.1. Model Interpretation in General Domain

Researchers attempt to interpret the decision of the model by calculating the importance score [19,33] and the heat map [18]. DeepLIFT [33] and KernelSHAP [19] compute the difference of scores between the original class and target class. However, both DeepLIFT and KernelSHAP are limited in the following ways. DeepLIFT has a limitation that users should find the reference input, which plays an important role in score calculation. In other words, the quality of the explanation depends on the experimenter's domain knowledge. KernelSHAP acquires the Shapley value, which is proposed to measure the feature impor-

tance in game theory [19]. Although authors introduce an approximate method to derive the Shapley value, their interpretation process still has a huge time cost [19]. In addition, Grad-CAM [18] creates a heat map, which is related with the class based on the feature map from the hidden layers. The heat map acts as the highlight of the most connected part of the input image. However, Grad-CAM has the limitation that, if there are multiple objects that are related with the class, it does not capture them.

### 2.2. Model Interpretation in Medical Domain

In the medical domain, researchers acquire the attention network and gradient to interpret segmentation models given images of magnetic resonance imaging (MRI), CT and skin lesions. SAUNet exploits the attention network to extract the global context of MRI segments [15]. Factorizer [16] solves non-negative matrix factorization as a self-attention role to segment brain tumors. The attention of both SAUNet and Factorizer is utilized to interpret the model. Moreover, gradient-based saliency maps are employed to interpret a skin cancer classifier [17]. Although they claim that their interpretations assure the decision of their model is similar to a human's, their interpretation methods are limited to the specific task, specifically the images.

Given the time series data in the medical domain such as electrocardiogram (ECG), NEVES [24] and LIMESegment [25], we find the most important subset of the input using LIME [20]. NEVES divides the signal by the same interval, while LIMESegment separates it by the change point, which is the most different point between window slides. Although they attempt to interpret the model given medical time series data, they are limited to only consider the aspect of time axis, whereas the frequency is also the important role to diagnose in practice.

## 3. Model Interpretation for Signal Classifier

As we mentioned in Section 2, model interpretation works have been proposed by many researchers. In the medical domain, most researchers focus on a model interpretation given image data, but few works deal with it given time series data. Specifically, in the medical domain, for diagnosis given time series data such as electroencephalogram (EEG) and electromyography (EMG), physicians consider not only periodic patterns, but also frequency bands [27–29], whereas existing methods take account of periodic patterns [24,25]. Therefore, we aim to develop an improved model interpretation given time series data. We simply define our problem as follows.

**Problem definition:** Given a black box classifier $f$, which is inaccessible model parameters, and to classify an input data $x$, which is a time series, our goal is to provide an interpretable representation $\xi(x)$ of the input.

In the rest of this section, we introduce our explanation method and the building blocks for our approach.

### 3.1. Prerequisites

LIME is an explanation method of machine learning models that is a black box classifier [20]. Given a model $f$ to be explained and an input instance $x$, LIME defines explanation $\xi(x)$ as follows using explainable model $g$, which has trainable parameters such as a linear model or decision tree:

$$\xi(x) = \arg\min_{g} \mathcal{L}(f, g, \pi_x) + \Omega(g) \tag{1}$$

where $\pi_x$ represent the distance between the input instance and the perturbed sample, as well as $\Omega(g)$ is the complexity of the explainable model $g$ such as the number of non-zero weights for a linear model. For the complexity $\Omega(g)$, we choose a constant so that LIME claims that $\Omega(g)$ can be a hyper-parameter [20].

Once the complexity is chosen, the input $x$ is divided by discrete components that are interpretable. For example, given an image of a dog in the grass and a complexity of 2, it

can be split into a set $\mathcal{Z}$ of two images such as the dog and the grass using a super-pixel such as Felzenszwalb [34]. Then, a trainable linear model $g$ is obtained by minimizing the loss as below:

$$\mathcal{L}(f, g, \pi_x) = \sum_{z, z' \in \mathcal{Z}} \pi_x(z) \big( f(z) - g(z') \big)^2 \tag{2}$$

where $z$ and $z'$ represent the perturbed sample and the discrete component of the input $x$ named perturbed sample, respectively, e.g., a perturbed sample $z$ can be one of the image of the dog or the image of the grass. At that time, the positive model parameter of $g$ describes the importance of the component because entries of $z'$ are 0 or 1 where those entries represent "absence" and "presence" of the component, respectively. Consequently, selection of the split method plays an important role. For time series data, former works apply LIME as well [24,25,35]. Unlikely for images, they not only focus on how to split the input, but also focus on how to generate perturbed regions. They acquire random split [24,35] or finding the most probable change points comparing the similarity between window slides [25]. In addition, they attempt to fill the regions to generate perturbed samples using constant value, linear interpolation or both short-time Fourier transform (STFT) and inverse STFT (ISTFT).

### 3.2. Perturbed Sample Generation for Time Series Data

Our approach is based on the given time series data being decomposed into two-dimensional information, which are periodic patterns on the time axis and frequency characteristics on the frequency axis. Moreover, as we mentioned earlier in Section 3.1, the selection of both split method and filling method are important to generate the most probable perturbed samples to exploit LIME for time series data. Our split method and filling method are described in the following sections.

#### 3.2.1. Separating Representative Component of Signal Data Using STFT and Super-Pixel

In practice, observed signals in medical centers are non-stationary, e.g., recorded EEG for diagnosis seizure. In other words, biomedical signals change frequency over time. Moreover, because physicians consider both periodic patterns and frequency bands simultaneously to diagnose a patient [27–29], we exploit STFT to transform the input signal.

STFT is considered to capture the change of frequency content over time, whereas the conventional discrete time Fourier transform (DTFT) averages frequency contents over time [36]. STFT given the input signal is defined as follows:

$$X[k, m] = \sum_{n=m}^{m+(N-1)} x[n] w[n-m] e^{-j\omega(n-m)} \tag{3}$$

where $x[n]$ is the observed data point at time $n$, $w[\cdot]$ is the window function given the window size and $\omega$ represents the angular frequency as well as $N$ is the number of samples. Therefore, transformed $X$ is represented as a 2-dimensional matrix that has the axes of the frequency and the time segment. STFT describes the magnitudes of the frequencies for the $m$-th segment. For example, we transform the input $x$ using STFT and visualize it as the spectrogram at the left diagram in Figure 2. As we can see in the second-row, the high magnitude of the low frequency is represented well around the 19th time segment.

Since our main purpose in this subsection is to divide the input signal into representative components, our next step is to segment the spectrogram. In this work, we utilize Felzenszwalb [34], which is proposed to segment an image into sub-regions. The intuition of Felzenszwalb is that pixels in an image and the differences between pixel values are considered as nodes $v_i \in V$ and weights $e_{ij} \in E$ for nodes $v_i$ and $v_j$ in an undirected graph $G = (V, E)$. $v_i$ is already in a region $S$, but neighbor node $v_j$ is not yet. It compares the weights $e_{ij}$ to the maximum weight $Max(w)$ in the region $S$. After that, it decides whether those pixels are in a same region or not using threshold $\tau$. The threshold is added to

the maximum weight $Max(w)$. We apply it for the spectgrogram as shown in Figure 2. Consequently, we separate clusters, which are divided by associated frequency and time pixels as we depicted in the third-row of the left diagram.

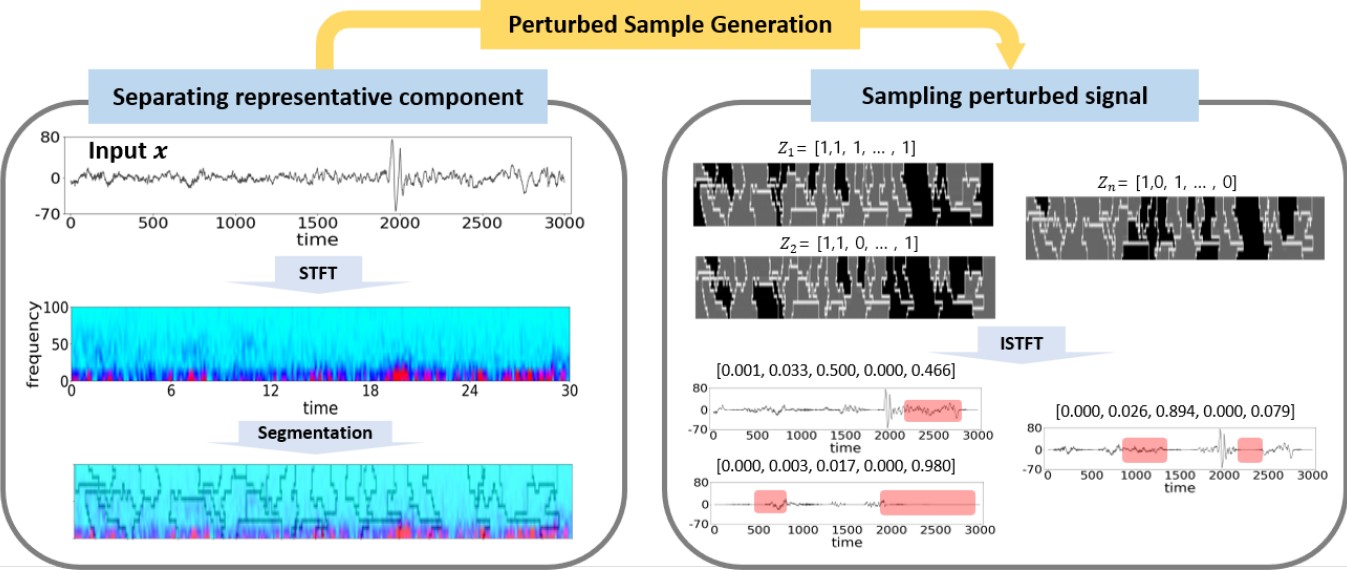

**Figure 2.** The process of generating perturbed samples consists of two sub processes: (1) As shown in the left diagram, given one-dimensional EEG signal $x$, the input is transformed using STFT. After that, it is divided using image segmentation such as Felzenszwalb [34]. (2) The divided samples are randomly permuted to generate perturbed samples as depicted at the upper side in the right diagram. Then, the perturbed samples are converted into one-dimensional signal again using ISTFT.

### 3.2.2. Restoration the Signal Data from Separated Representations

We generate a perturbed sample from the segmentation mentioned in Section 3.2.1. Because of the characteristics of Felzenszwalb, it divides around 40 segments. We select the masking ratio for the the predicted labels by the model $f$ to be explained as evenly distributed. More concretely, for an input instance $x$, we generate 100 masked samples for each masking ratio in range from 0.1 to 0.9 and choose the ratio, which maximizes the entropy of the prediction by the model $f$. In Figure 2, in the right diagram, we show the masking and sampling process. The upper three boxes are masked samples. Based on the masking regions of the spectrograms, we zero out the real and imaginary value obtained by STFT. Finally, we derive ISTFT as the following formula:

$$x[n] = \frac{1}{NW} \sum_{m=n-(N-1)}^{n} \sum_{k=0}^{N-1} X[k,m]e^{j\omega(n-m)} \tag{4}$$

where $W$ is the summation of the window function for all segments and other terms are same as Equation (3). In Figure 2, in the lower boxes in the right diagram, we depict the results of ISTFT. The inverse signals that are highlighted in red show that our approach masks the signal, not only the aspect of time but also the aspect of frequency.

### 3.3. Data Description

Our target model, which is to be explained, is trained using SleepEDF [37]. The research consists of the cassette study and the telemetry study, which are able to discover the effect of nothing from healthy patients and an insomnia medication temazepam from patients who have mild difficulty falling asleep, respectively. Because our purpose focuses on model interpretation, not the effect of a medicine, we consider only the dataset from the cassette study. The dataset is obtained from around 40 healthy patients of varied ages and sexes. The original dataset includes various channels, but we use only one channel of EEG

in this work. Each record is around 8 h. We split the record for each epoch, which is 30 s. Moreover, the dataset is annotated into six sleep stages, which are wakefulness, non-REM 1–4 and REM according to R&K rule [38], but we merge non-REM 3 and 4 as stated in the American Academy of Sleep Medicine (AASM) standard [26]. Therefore, the target model classifies five sleep stages given a 30 s EEG signal. Finally, we obtain 6955, 2581, 15951, 5051 and 7042 records of wakefulness, non-REM 1–3 and REM, respectively. We split the data into 75% and 25% to train and interpret the target model, respectively.

## 4. Experiments

In this section, we describe our experimental environment and results as well as we discuss our results. Every method including ours is implemented using PyTorch 1.11 in Python 3.8. All experiments reported in this section are performed on machines with Intel(R) Core(TM) i9-7900X CPU @ 3.30 GHz and 128 GB main memory running Ubuntu 18 OS. We also utilize a single GPU and NVIDIA GeForce 1080 Ti equipped with 11 GB of memory.

### 4.1. Implementation Methods

We implement a target model, which is to be explained, and the interpretation methods in the following list.

- **Target model:** Since our main goal is to interpret a deep learning classifier, we train a deep learning model, which is to be explained, by the proposed method. The target model is proposed for automatic sleep stage classification [39]. We train the model using the SleepEDF dataset, which is described in Section 3.3. The model consists of convolutional neural networks (CNNs) and long short-term memory networks (LSTMs). CNNs and LSTMs in the model are designed to capture features from a given epoch of an electroencephalography (EEG) signal and from sequential epochs of EEG signals, respectively. Because only CNN layers, which are called FeatureNet, achieve comparative performance with the full model in [39], our target model is FeatureNet.
- **LIMESegment [25]:** We compare our method with LIMESegment. They also exploit LIME to interpret a classifier for time series. They focus on detecting the change point given the time series data, which are considered the boundary of segments. Once the change points of the input signal are determined comparing the similarity of window slides, it generates perturbed samples. To create the perturbed samples, the intuition is that the segment in the input signal is filtered based on the threshold frequency. The threshold is determined by the highest frequency value over time with minimal variance. After that, anything lower than the threshold is filtered out. We exploit the imeplementation given by the authors from their GitHub repository.
- **Ours:** We also implement our interpretation method, which is described in Section 3.

### 4.2. Qualitative Results

Since we aim to provide the interpretable representation, we measure how much the representation describes the decision of the target model given the input well. Therefore, we remove the most representative region searched by our work as well as LIMESegment, the baseline. Then, we compute the difference of the logit. The experiment is based on our assumption as follows. A modified input that is removed from the most representative region is given to the target model. Then, we assume that the logit will be significantly changed.

We remove 25% of the most representative region according to each method and calculate the difference of the logit between the modified input and associated original input. For example, let a signal of wakefulness given to the target model produce the logit 0.85. If we remove 25% of the most representative region in the signal and it is given to the target model, the logit will be decreased to some value such as 0.35. Then, the difference of the logit is 0.5. In other words, the representative region provides probable decision criteria of the target model given the signal. We list the mean of the difference for the sleep stages, which are wakefulness, non-REM 1–3 and REM in Table 1. In the table, the five

sleep stages are annotated as "W", "N1", "N2", "N3" and "REM". In the table, the mean of the difference of logits shows that our work finds a more representative region than the baseline.

Subsequently, we exploit *t*-test for each class, which compares the means of two groups and whether they are significantly different or not. We set the null hypothesis and the alternative hypothesis for the test as below.

- **Null hypothesis:** The representations obtained by ours and LIMESegment are same.
- **Alternative hypothesis:** The representation obtained by ours is more critical for the model decision than the baseline.

Because the standard deviations are not same between ours and the baseline, we apply Welch's *t*-test and set alpha 10%. Finally, we obtain *p*-values for each sleep stage less than $1 \times 10^{-5}$. Therefore, we reject the null hypothesis and statistically prove that our method is significantly critical to describe the model decision.

**Table 1.** The difference of the logit between the input, which is removed from the most representative region and associated original input. The entry describes the mean of the the difference with regard to the sleep stage. Each column is for wakefulness, non-REM 1–3 and REM.

| Methods\Sleep Stages | W | N1 | N2 | N3 | REM |
|---|---|---|---|---|---|
| LIMESegment | 0.07 | 0.23 | 0.04 | 0.24 | 0.05 |
| Ours | 0.46 | 0.76 | 0.51 | 0.99 | 0.65 |

*4.3. Case Study*

According to the AASM manual [26], the K-complexes are shown during the sleep stage of non-REM 2 and they are considered strong evidence for the sleep stage. A K-complex is a wave starting from a high-voltage peak and followed by slow and large negative complex peaks around 350–550 ms. Moreover, there is an important clue for the sleep stage of non-REM 3. If the delta wave exists for more than 50% in an epoch, it is considered N3 [26]. The delta wave is the frequency between from 0.4 to 5 Hz. In this section, we show samples of representations obtained by ours and the baseline. Then, we discuss how the samples are representative, such as the rule stated in the AASM manual.

In Figure 3a, we depict the representations obtained by us and LIMESegment for N2. The representative regions are filled with light green. The 1st and 4th rows show the most representative regions for each method. Note that ours is selected from the spectrogram transformed by STFT while the baseline is raw signals. The most representative region for the class is shown as the blue lines in the 2nd and 5th rows, whereas without the representation is shown as the red lines in 3rd and 6th rows. The title of each signal explains the predicted class from the target model given the signal. As shown in the figure, the proposed method captures the K-complexes well, as pictured in the blue boxes. The baseline captures the K-complexes as the most opposite representation in the red boxes. Therefore, the only representation signal obtained by ours is still classified as the original class N2 as shown by the blue lines. However, the representation from the baseline is not classified as the original class, rather it is classified as N1 the blue line at the upper sample. Moreover, without the representative region, both signals from ours are misclassified as REM, but from the baseline is still classified as N2, the red line at the lower sample. In addition, when we retain only the most opposite representation by ours, the remaining signals are classified as different sleep stages as REM and N1, while the baseline ones are classified as the original sleep stage.

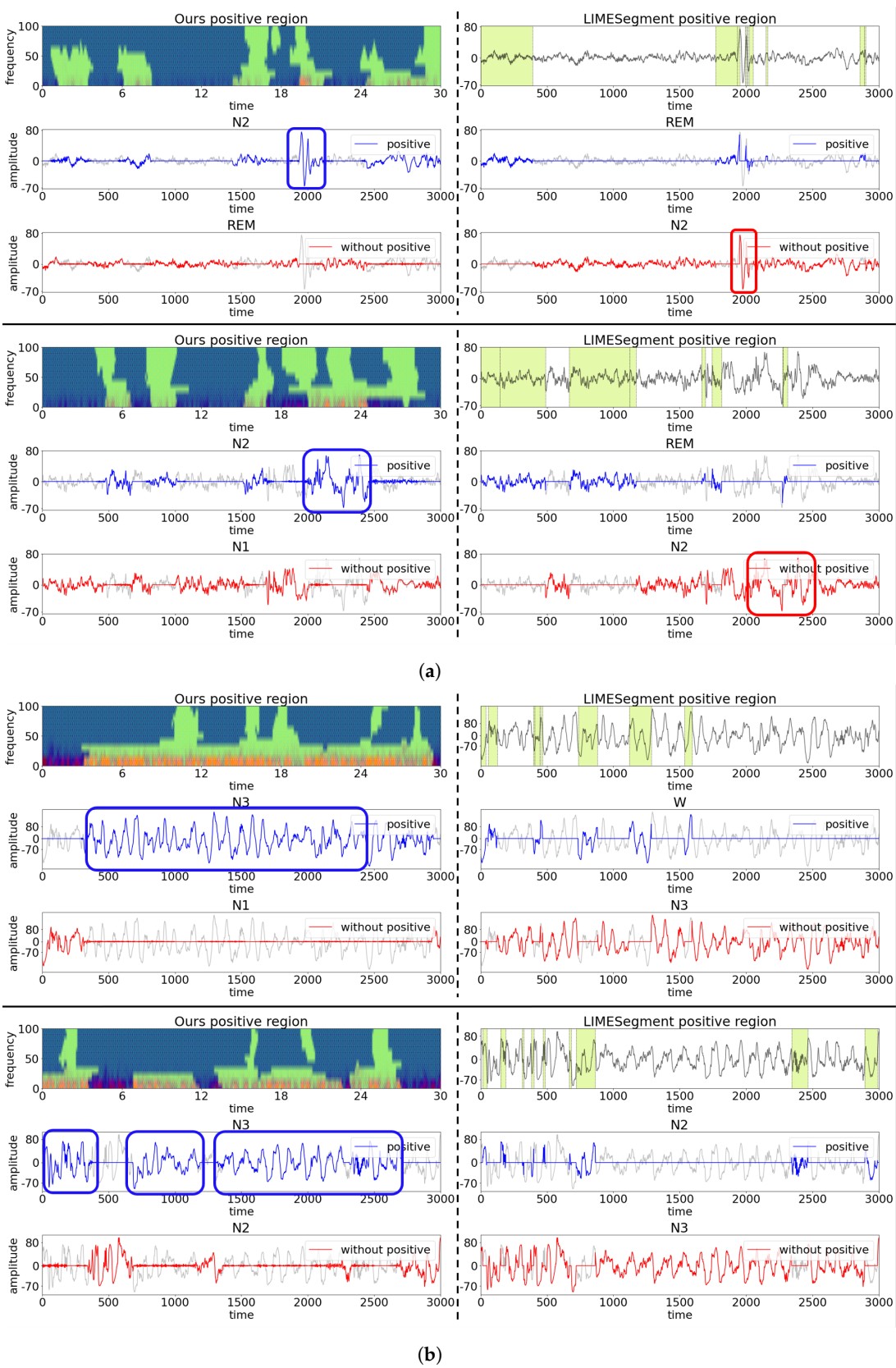

**Figure 3.** We depict how the representations are critical for the decision of the target model. The blue box and the red box indicate the most probable positive explanation of the target model and the most probable negative explanation of the target model given the signal respectively. They can be acquired to support physicians to interpret the target model. (**a**) Original sleep stage of the EEG samples is N2. (**b**) Original sleep stage of the EEG samples is N3.

In addition, we also draw the representative regions from both methods in Figure 3b. The regions are filled light green as well. As we can see in the figure, our method captures delta waves well in the blue boxes at the 2nd and 5th rows, whereas the baseline does only part of it. Therefore, only representative signals from ours are classified as the original sleep stage N3, but the baseline ones are misclassified as N2. Without the representation, the signals made by ours are misclassified as N1 and N2, while the signals from the baseline are still classified as N3.

## 5. Discussion

In this section, we analyze the results produced in the experiments. Moreover, we introduce a the possible scenario of developing applications using our method in practice. In addition, we discuss the limitations of our method and the future direction.

### 5.1. Analysis of Results

Our method is confirmed to outperform the baseline as described in Section 4.2. It is proved using statistical analysis. Moreover, in Section 4.3, it shows that the explanations obtained by ours for sleep stage N2 and N3 represent plausible reasons of diagnosis by the target model as physicians, as stated in the AASM standard [26]. Specifically, our method captures not only the explanation of the K-complexes for sleep stage N2, but also the explanation of delta waves for sleep stage N3. We analyze that because our method considers more fine-grained components of the input signal, and unlike the baseline considers coarse components, it is available to capture more representative regions.

### 5.2. Possible Application of Proposed Method

Moreover, our method can be adopted to develop an application to support physicians who want to utilize deep learning-based computer-aided diagnosis (CAD). For example, a physician can reduce the time cost of diagnosis using the CAD, and the reliability of the CAD is provided using ours, such as the explanations as depicted as the blue lines in Figure 3.

### 5.3. Limitations and Future Directions

Although our method considers both time and frequency axes together, it is limited in that the suggestion in this work depends on a clustering method for a transformed signal such as Felzenszwalb [34], which is exploited in this work. Moreover, the clustering method could not take account of the important bands of frequency. For example, according to the AASAM manual [26], humans consider the frequency at most beta waves, which is between 14–30 Hz. However, the clustering method does not consider the bands. Therefore, our method captures the over-explained sample from 0–100 Hz as shown in the region filled with light green in Figure 3. In the future, we plan to elaborate our method to consider the frequency more precisely and apply the model given the signal, not only EEG, but also different ones such as ECG or EMG.

## 6. Conclusions

In this work, we introduced the method to generate perturbed samples considering time and frequency axes to interpret the model given the signal data acquiring LIME. In the experiments, we provided our model interpretation, which captures significantly more important representation compared to the state-of-the art in a statistical manner. Moreover, in the case study, we showed that the representation obtained by ours reflects the physician's standards, such as AASM. We expect that our method promotes the model based on deep learning and can support computer-aided diagnosis in medical centers.

**Author Contributions:** Conceptualization, W.L. and Y.K.; methodology, W.L.; software, W.L., G.K. and J.Y.; validation, W.L., G.K. and J.Y.; formal Analysis, W.L.; investigation, W.L., G.K. and J.Y.; data curation, W.L., G.K. and J.Y.; writing-original draft preparation, W.L., G.K. and J.Y.; writing-review and editing, all authors; visualization, G.K. and J.Y.; supervision, W.L. and Y.K.; project administration, W.L. and Y.K.; funding acquisition Y.K. All authors have read and agreed to the published version of the manuscript.

**Funding:** This work was supported by the National Research Foundation of Korea (NRF) grant funded by the Korea government (MSIT) (No. 2020R1G1A1011471) and the Artificial Intelligence Convergence Research Center (Hanyang University ERICA), under Grant 2020-0-01343.

**Institutional Review Board Statement:** Not applicable.

**Informed Consent Statement:** Not applicable.

**Data Availability Statement:** Restrictions apply to the availability of these data. Data was obtained from "Sleep-EDF Database Expanded" and are available https://www.physionet.org/content/sleep-edfx/1.0.0/ (accessed on 1 July 2022).

**Conflicts of Interest:** The authors declare no conflict of interest. The funders had no role in the design of the study; in the collection, analyses, or interpretation of data; in the writing of the manuscript, or in the decision to publish the results.

## Appendix A. Additional Representations

In this section, we attach additional representations obtained by our work and LIME-Segment. In Figure A1, our representations show that it captures K-complexes well, which is the hard evidence to annotate as N2, while the baseline is not. In addition, we depict how ours catches delta waves, which is the important clue for N3, whereas the baseline only does partly in Figure A2.

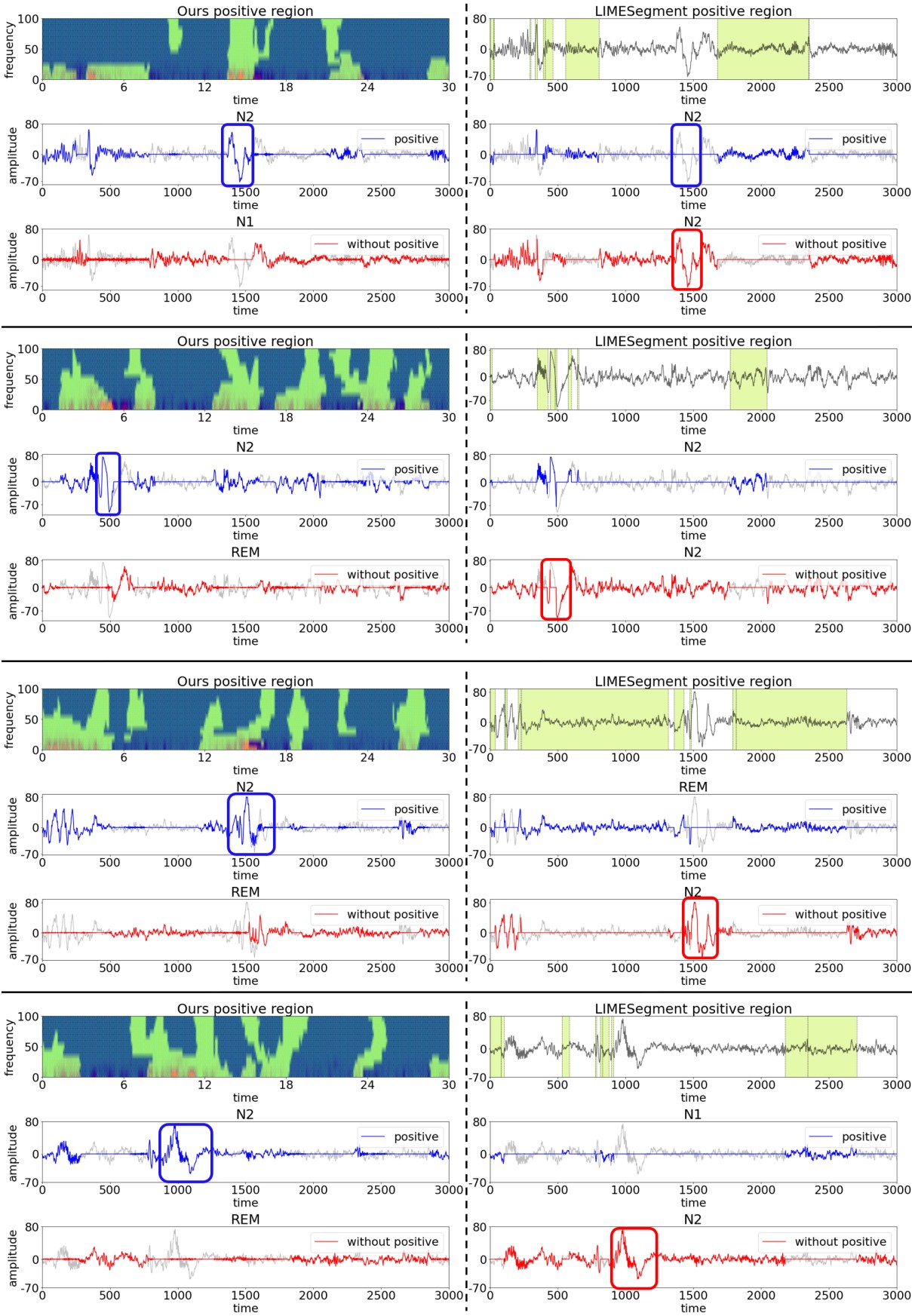

**Figure A1.** Additional representations obtained by ours and LIMESegment for N2.

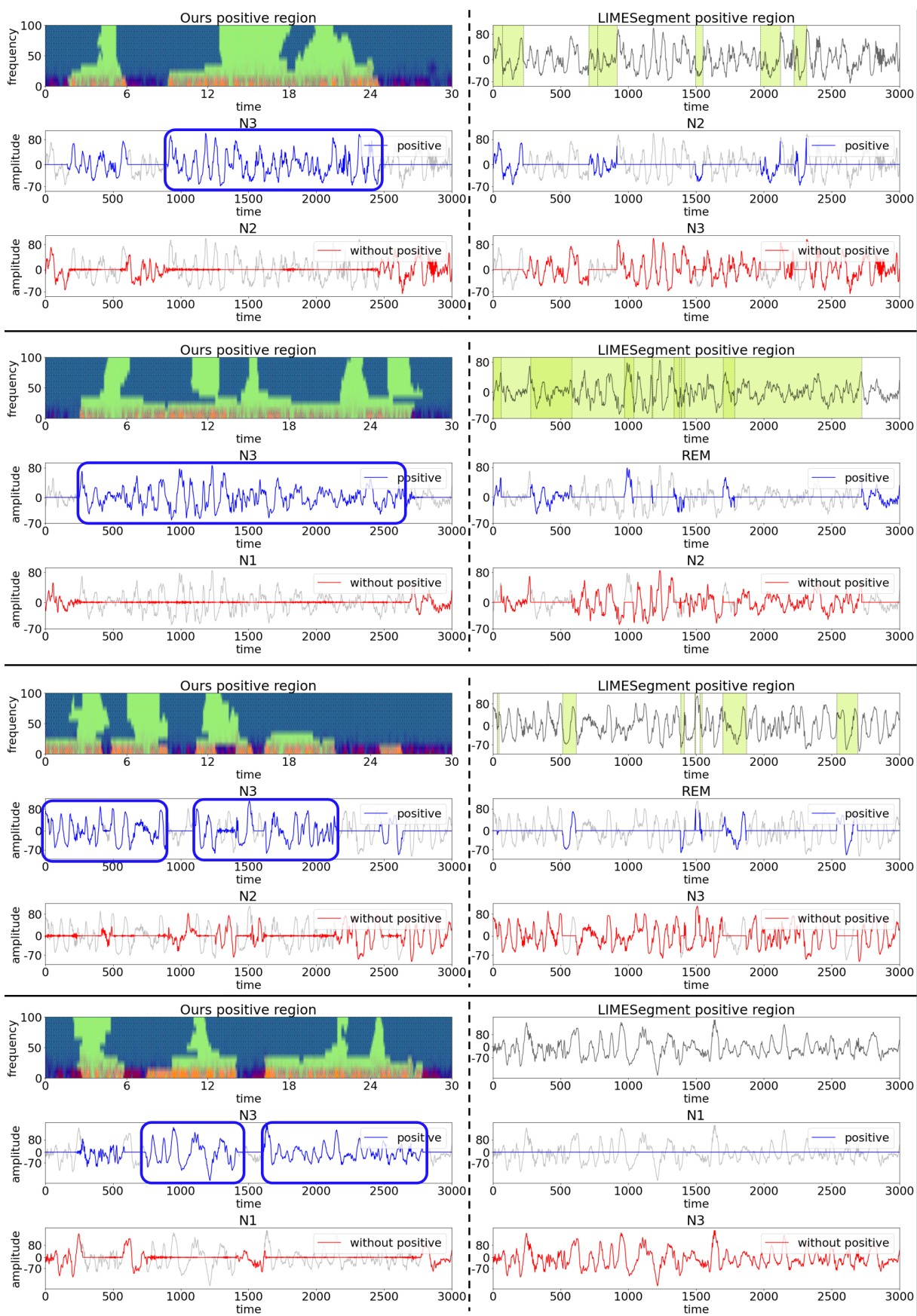

**Figure A2.** Additional representations obtained by ours and LIMESegment for N3.

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
