# Peer review of "Model Interpretation Considering Both Time and Frequency Axes Given Time Series Data"

_applsci, doi:10.3390/app122412807_

Round 1

Reviewer 1 Report

Please find the attachment

Reviewer 2 Report

In this manuscript entitled: “Model Interpretation considering both Time and Frequency Axes given Time Series Data” Woonghee Lee et al., nicely demonstrate a new method for deep learning model. Using both Time and Frequency Axes, they are showing that this new method is more relevant for medical clinical researchers.

The introduction and methods are really well written and easy to understand.

The conclusions are consistent with the results.

This article is suitable for publication in the journal.

Author Response

We are very appreciate for your careful review.

Reviewer 3 Report

Congratulations for your interesting work, thank you for detailing the methodology and for working on the quality/dimensions of the figures respecting the Template of the journal.

In the medical field, for time series data, existing interpretations consider only the time axis while physicians consider the frequency as well. In this work, the authors propose an interpretation of the model using LIME by considering the time and frequency axes.

The subject of research is interesting, especially its application in the medical field constitutes a contribution of new technologies in the benefit of health and medicine

Deep learning-based models have emerged in the medical field and are difficult to apply directly in practice and are considered unreliable.
This work focuses on the interpretation of ECG patterns using LIME offering explanations that originate from the permutation of split input data.
Hence the goal is to improve the performance of the interpretation which is highly dependent on the splitting method.

The authors have presented a well-designed working model, the only remark concerns the explanation of the methodology by adding some details on the interpretation approaches considered based on the division methods.

Conclusion and the references are presented in appropriate way.

Round 2

Reviewer 1 Report

Please remove the citations inside the conclusion. 

   I am accepting the manuscript. But remove it in the final version. 
